

# Data-driven remaining useful life prediction based on domain adaptation

Bin cheng Wen, Ming qing Xiao, Xue qi Wang, Xin Zhao, Jian feng Li and Xin Chen

ATS Lab, Air Force Engineering University, Xi'an, Shanxi, China

## ABSTRACT

As an important part of prognostics and health management, remaining useful life (RUL) prediction can provide users and managers with system life information and improve the reliability of maintenance systems. Data-driven methods are powerful tools for RUL prediction because of their great modeling abilities. However, most current data-driven studies require large amounts of labeled training data and assume that the training data and test data follow similar distributions. In fact, the collected data are often variable due to different equipment operating conditions, fault modes, and noise distributions. As a result, the assumption that the training data and the test data obey the same distribution may not be valid. In response to the above problems, this paper proposes a data-driven framework with domain adaptability using a bidirectional gated recurrent unit (BGRU). The framework uses a domain-adversarial neural network (DANN) to implement transfer learning (TL) from the source domain to the target domain, which contains only sensor information. To verify the effectiveness of the proposed method, we analyze the IEEE PHM 2012 Challenge datasets and use them for verification. The experimental results show that the generalization ability of the model is effectively improved through the domain adaptation approach.

## INTRODUCTION

Prognostics aims to provide reliable remaining useful life (RUL) predictions for critical components and systems *via* a degradation process. Based on reliable forecast results, managers can determine the best periods for equipment maintenance and formulate corresponding management plans; this is expected to improve reliability during operation and reduce risks and costs. Typically, prognostic methods are classified into model-based methods and data-driven methods (*Heng et al., 2009*).

Model-based methods describe the degradation process of engineering systems by establishing mathematical models based on the failure mechanism or the first principle of damage (*Cubillo, Perinpanayagam & Esperon-Miguez, 2016*). However, the physical parameters of the model should vary with different operating environments, so the uncertainty of parameters limits the application of such methods in industrial systems (*Pecht & Jaai, 2010*).

Corresponding author
Bin cheng Wen, 1255292204@qq.com

Different from model-based methods, data-driven methods can construct the mapping relationship between historical data and RUL information but do not need to study the degradation mechanism of the given system. Data-driven methods have become the focus of research due to their powerful modeling capabilities. Among them, machine learning, as a very common data-driven method, has been widely used in the field of RUL prediction. For example, Theodoros H proposed an E-support vector machine (SVM) method to predict the remaining life of a rolling bearing (*Loutas, Roulias & Georgoulas, 2013*). To solve the limitations of SVMs, Wang proposed an RUL prediction method based on a relevance vector machine (RVM) (*Wang, Youn & Hu, 2012*). Selina proposed a naive Bayes-based RUL prediction model for lithium-ion batteries (*Ng, Xing & Tsui, 2014*). Wu used a random forest (RF) to predict tool wear (*Wu et al., 2017*). However, machine learning methods require manual extraction or signal processing and statistical projection to obtain health factors. On the other hand, feature extraction is separate from parameter training.

Recently, deep neural networks have been widely used in the field of RUL prediction due to their powerful feature extraction capabilities and regression analysis capabilities (*Zhu, Chen & Peng, 2019*; *Li, Ding & Sun, 2018*; *Deutsch & He, 2018*; *Wu et al., 2018*). Deep learning not only combines feature extraction with the parameter training process but can also automatically learn relevant features instead of manually designing them. This greatly compensates for the shortcomings of machine learning. At the same time, most of the signals collected by the associated sensors are time series. Some deep neural networks that can support sequence data as inputs are also widely used in RUL prediction. For example, recurrent neural networks (RNNs) (*Malhi, Yan & Gao, 2011*; *Heimes & Ieee, 2008*; *Gugulothu et al., 2017*), long short-term memory networks (LSTMs) (*Zhang et al., 2018*), and gated recurrent units (GRUs) (*Lu, Hsu & Huang, 2020*) are common approaches.

Although data-driven methods based on deep learning have achieved good results on RUL prediction tasks, in such methods, the network needs to be trained with a large number of labeled datasets to obtain a sufficiently accurate model. However, for complex systems, it is often difficult to collect sufficient data with run-to-failure information. Furthermore, the current methods based on deep learning require the training data and the test dataset to follow similar distributions, which means that the dataset needs to come from the same feature space. However, in the actual application process, due to the changing environment in which equipment operates, differences in data distribution are widespread, which leads to a decline in RUL prediction accuracy in actual applications. In other words, the RUL prediction model obtained through the training dataset may not have good generalization ability, and the performance on the test dataset may be poor.

To solve these problems, some domain adaptation methods have been designed gradual application in the field of prognostics and health management (PHM). The purpose of domain adaptation is to use existing knowledge to solve different but related problems. This means that we can use a number of models trained with labeled data to adapt to data with different input characteristics, data distributions, and limited or no labeled data. However, the existing life prediction method based on domain adaptation is difficult to adapt to multivariate sensor data because it does not consider the temporal dependency

problem (*Da Costa et al., 2020*). As a result, the existing RUL methods based on transfer learning (TL) can hardly adapt to common RUL prediction problems.

In this article, we propose the use of bidirectional GRUs (BGRUs) to solve the problem of sequential data processing. We use labeled source domain data and unlabeled target domain data for training. This can be viewed as a process of unsupervised learning based on feature transfer. At the same time, we use a domain-adversarial neural network (DANN) to learn features with domain invariance. To verify the method proposed in this article, we use the IEEE PHM 2012 Challenge datasets for verification. The experimental results prove the effectiveness of the method proposed in this article.

The main contributions of our work are as follows:

(1) We propose a new RUL prediction structure that can better adapt to data distribution shifts under different working environments and fault modes.

(2) The framework not only uses a single sensor but also integrates information from multiple sensors.

(3) Compared with the nonadaptive method and the traditional nondeep adaptive method, our proposed structure obtains better prediction results.

The rest of this article is organized as follows: 'Literature Review' briefly introduces the theoretical background of TL and deep learning. Then, the experimental procedure is introduced in 'Materials'. In 'Methods', the BGRU, DANN, domain-adaptive BGRU and BGRU-DANN structures proposed in this article are introduced. On this basis, RUL prediction for a bearing dataset is studied. The comparative results and conclusions are given in 'Results'.

## LITERATURE REVIEW

### Deep learning and PHM

Within the framework of deep learning, RNN is a very representative structure. It can not only process sequence data but also extract features well. Furthermore, RNNs have been used in the field of RUL prediction (*Yu, Kim & Mechefske, 2020*; *Guo et al., 2017*). However, such networks cannot deal with the weight explosion and gradient disappearance problems caused by recursion. This limits their application in long-term sequence processing. To solve this problem, many RNN variants have begun to appear, for example, LSTM and GRUs. These networks can process series with long-term correlations and extract features from them.

As a variant of the RNN proposed earlier, LSTM has already performed well in RUL prediction. *Shi & Chehade (2021)* also showed similar results; real-time, high-precision RUL prediction was achieved by training a dual-LSTM network. *Chen et al. (2021)* tried to add the attention mechanism commonly used in the image field to an LSTM network and proposed an attention-based LSTM method, which also achieved good results. *Ma & Mao (2021)* proposed integrating deep convolution into the LSTM network. This approach applies a convolution structure to output-to-state and state-to-state information and uses time and time-frequency information simultaneously.

As another type of RNN variant, GRUs have also begun to be applied in RUL prediction. Compared with LSTM, a GRU has a simpler structure and fewer parameters, but the effect

is comparable to that of LSTM. *Deng et al. (2020)* combined a GRU with a particle filter (PF) and proposed an MC-GRU-based fusion prediction method, which achieved good performance in a prognostic study of ball screws. *Lu, Hsu & Huang (2020)* proposed a GRU network based on an autoencoder. It uses an autoencoder to obtain features and a GRU network to extract sequence information.

Compared with standard unidirectional LSTM and GRU, the bidirectional structure can extract better feature information (*Yu, Kim & Mechefske, 2019*)Huang proposed to combine multi-sensor data with operation data to make RUL prediction based on bidirectional LSTM (BLSTM) (*Huang, Huang & Li, 2019*). *Huang et al. (2020)* proposed a fusion prediction model based on BLSTM. It not only proves the advantages of LSTM in automatic feature acquisition and fusion, but also demonstrates the excellent performance of BLSTM in RUL prediction. *Yu, Kim & Mechefske (2019)* proposed a Bidirectional Recurring Neural Network based on autoencoder for C-MAPSS RUL estimation. She attempted to use BGRU for RUL prediction and validated its effectiveness with Bearing data (*She & Jia, 2021*). There are other deep networks, such as CNN, that are also widely used in the PHM space (*Wang et al., 2021*).

## Transfer learning

In most classification or regression tasks, it is assumed that sufficient training data with label information can be obtained. At the same time, it is assumed that the training data and the test data come from the same distribution and feature space. However, in real life, data offset is common. The training data and test data may come from different marginal distributions. As a way to find the similarity between the source domain and the target domain, TL has achieved good results in domain adaptation. The basic TL methods can be divided into the following categories:

(1) Instance-based TL;
(2) Feature-based TL;
(3) Model-based TL;
(4) Relation-based TL.

Detailed information about these methods can be found in the literature (*Pan & Yang, 2010*). In this article, they are divided into two categories according to their development process. One contains nondeep learning methods, and the other is based on deep learning methods.

The most representative nondeep learning approaches are a series of methods based on maximum mean discrepancy (MMD). For example, *Pan et al. (2011)* proposed transfer component analysis (TCA), which is the most representative TL method. Long tried to combine marginal distributions and conditional distributions and proposed joint distribution adaptation (JDA) (*Long et al., 2013*). *Wang et al. (2017)* believed that marginal distributions and conditional distributions should have different weights. As a result, he proposed balanced distribution adaptation (BDA). This technique minimizes the distance between the source domain and the target domain through feature mapping so that the data distributions of the two domains can be as similar as possible. There are also some other nondeep learning methods. For example, *Tan & Wang (2011)* proposed structural

correspondence learning (SCL) based on feature selection. Sun and Gong proposed correlation alignment (CORAL) (*Sun & Saenko, 2016*) and the geodesic flow kernel (GFK) method (*Gong et al., 2012*) based on subspace learning.

With the continuous development of deep learning methods, an increasing number of people are beginning to use deep neural networks for TL. Compared with traditional nondeep TL methods, deep TL has achieved the best results at this stage. The simplest method for conducting deep TL to finetune the deep network, which realizes transfer by finetuning the trained network (*Razavian et al., 2014*). At the same time, by adding an adaptive layer to deep learning, deep network adaptation has also begun to appear consistently. For example, Tzeng proposed deep domain confusion (DDC) (*Tzeng et al., 2014*), *Long et al. (2015)* proposed a domain adaptive neural network, *Long et al. (2017)* proposed a joint adaptation network (JAN), etc. Recently, as the latest research result in the field of artificial intelligence, generative adversarial networks (GANs) have also begun to be used in transfer learning. *Ganin et al. (2017)* first proposed the DANN. *Yu et al. (2019)* extended a dynamic distribution to an adversarial network and proposed dynamic adversarial adaptation networks (DAANs).

## Transfer learning and PHM

As a way of thinking and a mode of learning, transfer learning has a core problem: finding the similarity between the new problem and the original problem. TL mainly solves the following four contradictions (*Yu et al., 2019*):

(1) The contradiction between big data and less labeling.
(2) The contradiction between big data and weak computing.
(3) The contradiction between a universal model and personalized demand.
(4) The needs of specific applications.

The above four contradictions also exist in PHM. For example, with the development of advanced sensor technology, an increasing amount of data have been collected. However, the amount available data with run-to-failure label information is still small. Second, because the operating state of equipment is affected by many different conditions, the data collected are often not representative due to the differences between various operating conditions and environments. Thus, it is difficult to construct a predictive model with strong universality. Finally, for a PHM system, because of the complexity of the object's use environment, we also need an RUL prediction model with specific applications. However, because there are no data with sufficient label information, it is impossible to use a data-driven approach to build an accurate predictive model. As an effective means, TL can help solve the existing problems of PHM. However, in the field of PHM, TL is mainly used in classification tasks (*Da Costa et al., 2020*). *Shao et al. (2020)* proposed a convolutional neural network (CNN) based on TL, which is used to diagnose bearing faults under different working conditions. *Xing et al. (2021)* proposed a distribution-invariant deep belief network (DIDBN), which can adapt well to new working conditions. *Feng & Zhao (2021)* pointed out that it is necessary to conduct fault diagnosis research with zero samples. They introduced the idea of zero-shot learning into industrial fields and proposed a zero-sample fault diagnosis method based on the attribute transfer method.

**Table 1  Test data information.**

| Pitch diameter | Diameter of the roller | Number of rollers | Contact angle |
|---|---|---|---|
| 25.6 mm | 3.5 mm | 13 | 00 |

RUL prediction studies based on TL are still relatively few in number, as far as the authors know (*Fan, Nowaczyk & Rögnvaldsson, 2020*; *Mao, He & Zuo, 2020*; *Sun et al., 2019*; *Zhu, Chen & Shen, 2020*).

# MATERIALS

## Experimental analysis

In this section, we first describe the experimental data and platform in detail. Then, we analyze the data processing and feature extraction methods and introduce the relevant performance metrics. Finally, the effectiveness of our proposed method is verified *via* a comparison with other methods.

## Experimental data description

The IEEE PHM Challenge 2012 bearing dataset is used to test the effectiveness of the proposed method. This dataset is collected from the PRONOSTIA test platform and contains run-to-failure datasets acquired under different working conditions.

PRONOSTIA is composed of three main parts: a rotating part, a degradation generation part and a measurement part. Vibration and temperature signals are gathered during all experiments. The frequency of vibration signal acquisition is 25.6 kHz. A sample is recorded every 0.1 s, and the recording interval is 10 s. The frequency of temperature signal acquisition is 10 Hz. 600 samples are recorded each minute. To ensure the safety of the laboratory equipment and personnel, the tests are stopped when the amplitude of the vibration signal exceeds 20 g. The basic information of the tested bearing is shown in Table 1. Table 2 gives a detailed description of the datasets. From the table, we can see that the operating conditions of the three datasets are different, and from the literature (*Zhu, Chen & Shen, 2020*), we can obtain that the failure modes are also different. This is very suitable for experimenting with the method proposed in this article. To verify the effectiveness of the method proposed in this paper, we divide the data into a source domain and target domain according to the different operating conditions. The basic information is shown in Table 3.

## Feature extraction

The original signal extracted by the sensor cannot reflect the degradation trend of the system well. At the same time, using original data for network training will increase the cost of network training and affect the final output result. It is necessary to extract the degradation information of the system by corresponding methods, which is called feature extraction.

From the raw vibration data, we extract 13 basic time-domain features. They are the maximum, minimum, mean, root mean square error (RMSE), mean absolute value, skewness, kurtosis, shape factor, impulse factor, standard deviation, clearance factor, crest

**Table 2  Descriptions of the experimental datasets.**

|  | Dataset 1 | Dataset 2 | Dataset 3 |
|---|---|---|---|
| Load (N) | 4000 | 4200 | 5000 |
| Speed (rpm) | 1800 | 1650 | 1500 |
| Dataset | bearing 1-1 | bearing 2-1 | bearing 3-1 |
|  | bearing 1-2 | bearing 2-2 | bearing 3-2 |
|  | bearing 1-3 | bearing 2-3 | bearing 3-3 |
|  | bearing 1-4 | bearing 2-4 |  |
|  | bearing 1-5 | bearing 2-5 |  |
|  | bearing 1-6 | bearing 2-6 |  |
|  | bearing 1-7 | bearing 2-7 |  |

**Table 3  Transfer prognostics task.**

| Transfer prognostics | Source | Target |
|---|---|---|
| Dataset 1-Dataset 2 | Labeled: bearing 1-3–bearing 1-7 | Unlabeled: bearing 2-1<br>Unlabeled: bearing 2-4<br>Unlabeled: bearing 2-6 |
| Dataset 1-Dataset 3 | Labeled: bearing 1-3–bearing 1-7 | Unlabeled: bearing 3-1<br>Unlabeled: bearing 3-2<br>Unlabeled: bearing 3-3 |

factor, and variance. At the same time, through 4-layer wavelet packet decomposition, we extract the energy of 16 frequency bands as time-frequency domain features. In the literature (*Zhu, Chen & Shen, 2020*), the frequency resolution of the vibration signal was too low. Therefore, we do not extract the frequency domain features but rather use the features of three trigonometric functions. They are the standard deviation of the inverse hyperbolic cosine (SD of the IHC), standard deviation of the inverse hyperbolic sine (SD of the IHS), and standard deviation of the inverse tangent (SD of the IT). For trigonometric features, trigonometric functions transform the input signal into different scales so that the features have better trends (*Zhu, Chen & Shen, 2020*), and the feature types are shown in Table 4. Through feature extraction, we can obtain 64 features from the feature dataset, which can better represent the degradation process of the system. Because of space constraints, we only show features along the $X$-axis of the bearing 1-1 data in Fig. 1.

## Data processing

By processing the original data, we extract a set of feature vectors, which are expressed as $X = (x_1, x_2, x_3, \ldots\ldots, x_N)$. To obtain a better experimental result, the experimental data need to be normalized. In this article, the maximum and minimum values are normalized, and the basic calculation formula is as follows:

$$\tilde{x}_t^{i,j} = \frac{x_t^{i,j} - \min(x^j)}{\max(x^j) - \min(x^j)} \tag{1}$$

where $x_t^{i,j}$ represents the *ith* value of the *jth* feature at the *tth* moment and $x^j$ is the *jth* inputted feature vector.

**Table 4  Feature set.**

| Type | Feature | |
| --- | --- | --- |
| Time-domain features | F1: Maximum | F8: Shape Factor |
| | F2: Minimum | F9: Impulse Factor |
| | F3: Mean | F10: Standard Deviation |
| | F4: RMSE | F11: Clearance Factor |
| | F5: Mean Absolute Value | F12: Crest Factor |
| | F6: Skewness | F13: Variance |
| | F7: Kurtosis | |
| Time-frequency domain features | F14-F29: Energies of sixteen bands | |
| Trigonometric features | F30: SD of the IHC | |
| | F31: SD of the IHS | |
| | F32: SD of the IT | |

## Sliding time window processing

After the extracted features are normalized, a sliding time window (TW) is used to generate the time series input $x^i = \{x_t^i\}_{t=1}^{T_\omega}$. The size of the input time window is $T_\omega$. The process of its generation is shown in Fig. 2:

## Performance metrics

We use three indicators to evaluate the performance of the proposed method. The mean absolute error (MAE), mean squared error (MSE) and R2_score provide estimations regarding how well the model is performing on the target prediction task. The formulas for their calculation are as follows.

MAE:

$$MAE = \frac{1}{L}\sum_{i=1}^{L}|y_i - \hat{y}_i|. \tag{2}$$

MSE:

$$MSE = \frac{1}{L}\sum_{i=1}^{L}(y_i - \hat{y}_i)^2. \tag{3}$$

R2_score:

$$R2\_score = 1 - \frac{\sum_{i=1}^{n}(y_i - \hat{y}_i)^2}{\sum_{i=1}^{n}(y_i - \overline{y}^2)^2}. \tag{4}$$

Here, $L$ is the length of the test data, $y_i$ is the ith true value, $\hat{y}_i$ is the corresponding predicted value, and $\overline{y}$ is the average of the true values.

## METHODS

### Problem definition

We use $T_S$ to denote the training task and $T_T$ to denote the target task. The training and testing data are represented as the source domain dataset $D_S$ and the target domain dataset $D_T$, respectively. $D_S = \{(x_S^i, y_S^i)\}_{i=1}^{N_s}$, where $x_S$ is a series of features belonging to the feature space, its length is $T_i$, and its characteristic number is $q_s$. $y_S$ represents

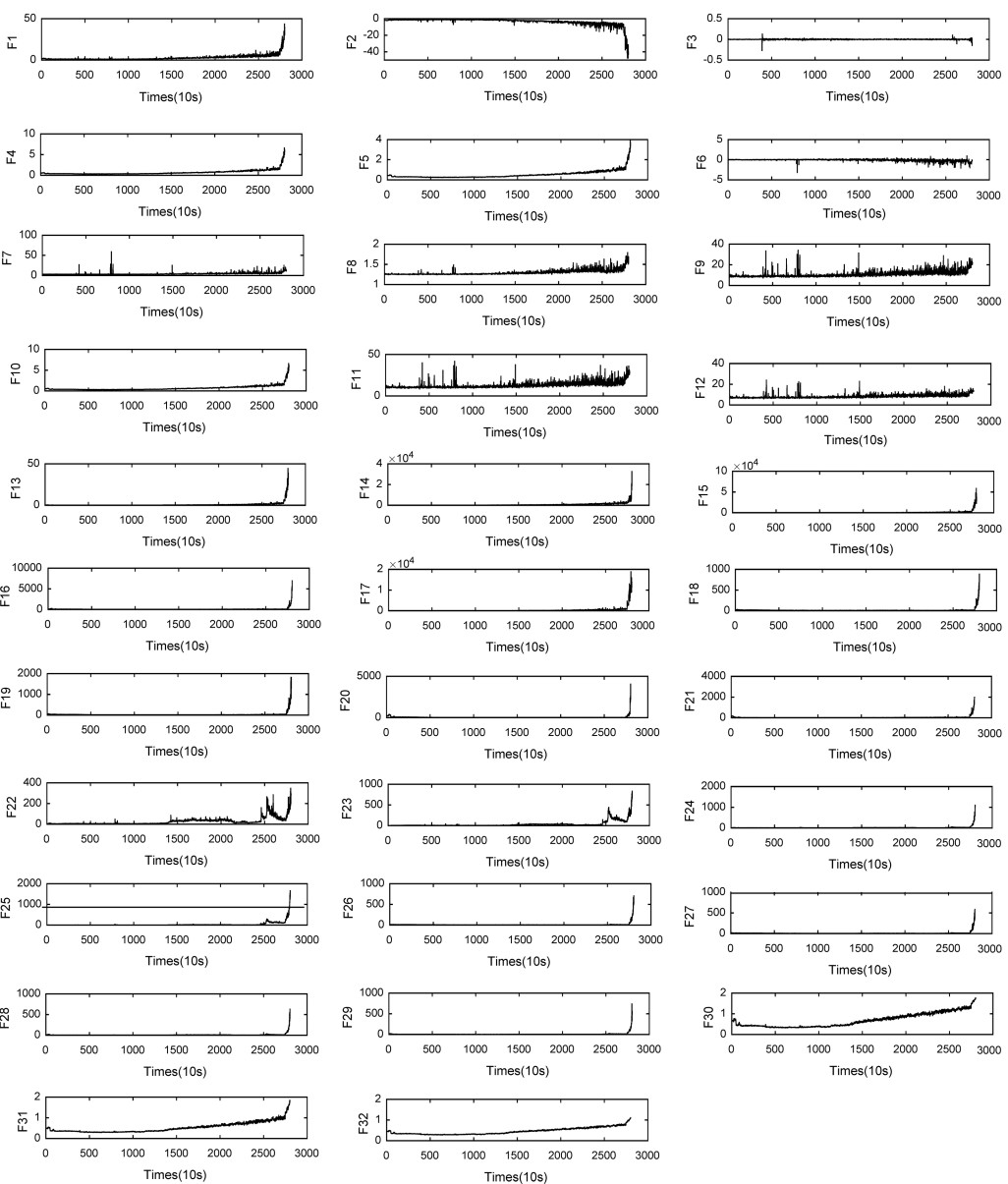

**Figure 1  Features for Bearing 1-1.**

the RUL label corresponding to the feature sequence $x_S$. $D_T = \{(x_T^i,)\}_{i=1}^{N_T}$, but it only contains characteristic information and no RUL information. We assume that the marginal probability distributions of $D_S$ and $D_T$ are not the same; that is, $P(X_S) \neq P(X_T)$. Here, we use source and target domain data to learn a prediction function $F$. The goal of the training process is to enable $F$ to estimate the corresponding RUL of the target domain samples during testing. During training, we use the corresponding datasets: $\{(x_S^i, y_S^i)\}_{i=1}^{N_s}$ from the source domain and $\{(x_T^i,)\}_{i=1}^{N_T}$ from the target domain. This is an unsupervised TL method. The process of training can be expressed as $y_S = F(x_S, x_T)$.

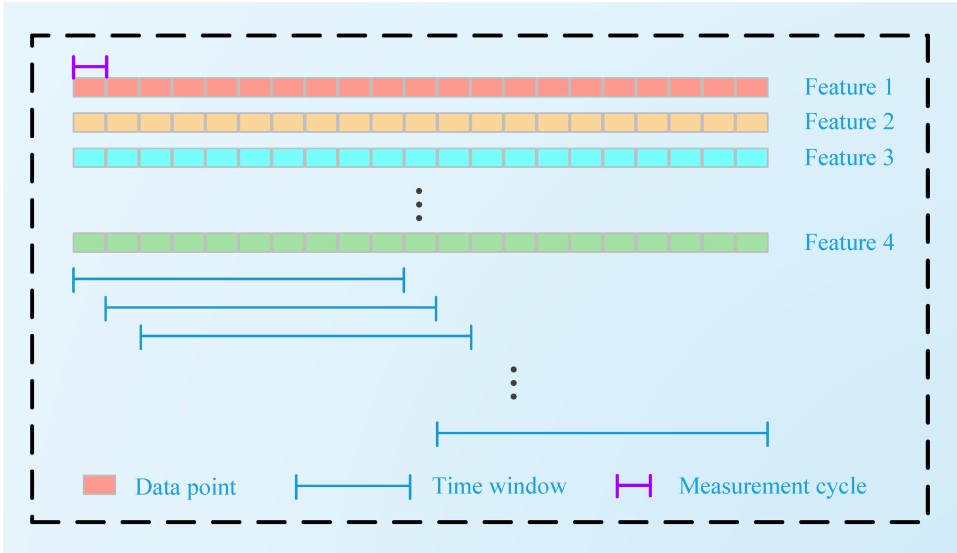

**Figure 2** Sliding TW processing technique.

### Bidirectional gated recurrent unit

A GRU is a variant of the LSTM structure. Compared with LSTM, its structure is simpler, and there are fewer parameters. He combined the forget gate and input gate in LSTM into a single update gate. At the same time, the cell state and hidden state were also merged. A GRU contains two door structures, a reset door and an update door. The reset gate determines whether the new input is combined with the output from the previous moment; that is, the smaller the value of the reset gate is, the less the output information from the previous moment is retained. The update gate determines the degree of influence of the output information from the previous moment on the current moment. The larger the value of the update gate is, the greater the influence of the output from the previous moment on the current output. The GRU-based structure is shown in Fig. 3.

In our proposed structure, a BGRU is used to obtain time series features from a TW $T_W$. Here, $x_t$ is the input at time $t$, and $h_t$ represents the output of the GRU at time $t$. $r_t$ is the reset gate, and $z_t$ is the update gate. These two parts determine how to obtain $h_t$ from $h_{t-1}$. The hidden layer of the GRU is defined as follows when running at time $t$: Forward propagation:

$$\overrightarrow{h}_t = f(\overrightarrow{x}_t, \overrightarrow{h}_{t-1}, \overrightarrow{\theta}_{BGRU}) \tag{5}$$

$$= \begin{cases} \overrightarrow{r}_t = \sigma(\overrightarrow{W}_r[\overrightarrow{h}_{t-1}, \overrightarrow{x}_t] + \overrightarrow{b}_r) \\ \overrightarrow{z}_t = \sigma(\overrightarrow{W}_z[\overrightarrow{h}_{t-1}, \overrightarrow{x}_t] + \overrightarrow{b}_z) \\ \overrightarrow{\tilde{h}}_t = \tanh(\overrightarrow{W}_h[\overrightarrow{r}_t \overrightarrow{h}_{t-1}, \overrightarrow{x}_t] + \overrightarrow{b}_h) \\ \overrightarrow{h}_t = (1 - \overrightarrow{z}_t)\overrightarrow{h}_{t-1} + \overrightarrow{z}_t \overrightarrow{\tilde{h}} \end{cases} \tag{6}$$

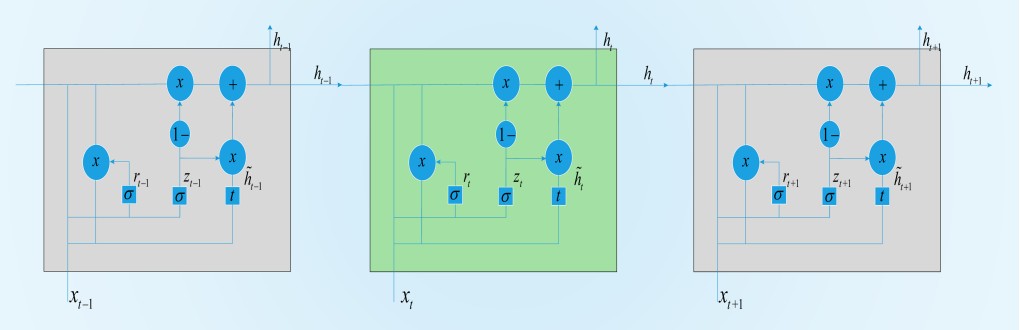

**Figure 3  GRU memory cell.**

Backward propagation:

$$\overleftarrow{h}_t = f(\overleftarrow{x}_t, \overleftarrow{h}_{t-1}, \overleftarrow{\theta}_{BGRU}) \tag{7}$$

$$= \begin{cases} \overleftarrow{r}_t = \sigma(\overleftarrow{W}_r[\overleftarrow{h}_{t-1}, \overleftarrow{x}_t] + \overleftarrow{b}_r) \\ \overleftarrow{z}_t = \sigma(\overleftarrow{W}_z[\overleftarrow{h}_{t-1}, \overleftarrow{x}_t] + \overleftarrow{b}_z) \\ \overleftarrow{\tilde{h}}_t = \tanh(\overleftarrow{W}_h[\overleftarrow{r}_t \overleftarrow{h}_{t-1}, \overleftarrow{x}_t] + \overleftarrow{b}_h) \\ \overleftarrow{h}_t = (1 - \overleftarrow{z}_t)\overleftarrow{h}_{t-1} + \overleftarrow{z}_t \overleftarrow{\tilde{h}}. \end{cases} \tag{8}$$

Here, $r_t$ controls how much information is passed to $h_t$, and its calculation is shown in Eqs. (6) and (8). $z_t$ determines the extent to which $h_{t-1}$ is passed to the next state, which is calculated as shown in the formula. $\rightarrow$, $\leftarrow$ represent the processes of forward and backward propagation, respectively. In the forward and backward propagation versions of the formula, $\sigma$ is the sigmoid activation function. $W_z$ is the update weight. $W_r$ is the reset weight. $b_r$ and $b_z$ are the deviations. $\tilde{h}_t$ indicates the candidate status. $h_t$ is the hidden layer output.

A portion of the input features of the BGRU can be expressed as $X_k = (x_{k,1}, x_{k,2}, \ldots\ldots, x_{k,t_{TW}})$. The output of the BGRU can be expressed as

$$H_k = [h_1, \ldots, h_t, \ldots, h_{t_{TW}}]$$
$$= f(X_k, \overrightarrow{\theta}_{GRU}, \overleftarrow{\theta}_{GRU}). \tag{9}$$

Here, $f(\cdot)$ represents the hidden layer function of the BGRU, as defined by Eqs. (6) and (8). $H_k = [h_1, \ldots, h_t, \ldots, h_{t_{TW}}]$ is the output characteristic. $(\overrightarrow{\theta}_{GRU}, \overleftarrow{\theta}_{GRU})$ represents the parameters of the forward and backward propagation operations. $h_t$ represents the output characteristics, and the formula for the base calculation is as follows:

$$h_t = \overrightarrow{h}_t \oplus \overleftarrow{h}_t. \tag{10}$$

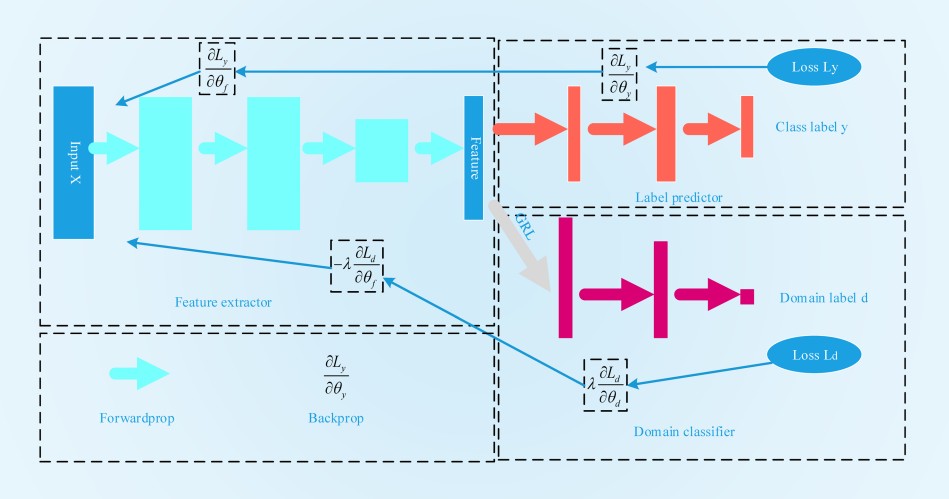

**Figure 4** **The flowchart of DANN.**

## Domain adversarial neural networks

Inspired by the GAN, *Ganin et al. (2017)* first proposed domain-adversarial training for neural networks, the process for which is shown in Fig. 4. A DANN combines domain adaptation with feature learning during the training process to better obtain distinctive and domain-invariant features. At the same time, the learned weights can also be directly used in the target field. The network structure of a DANN is mainly composed of three parts: a feature extractor $G_f$, a category predictor $G_y$ and a domain classifier $G_d$. To maximize the domain classification error, $G_f$ is used to extract the features with the greatest domain invariance. $G_y$ is used to classify the source domain data. $G_d$ is used to distinguish between the characteristic data of the source domain and the target domain. Its training objectives are mainly twofold: the first is to accurately classify the source domain dataset to minimize the category prediction error. The second is to confuse the source domain dataset with the target domain dataset to maximize the domain classification error. The loss function of the DANN can be expressed by the following formula:

$$L(\theta_f,\theta_y,\theta_d) = \sum_{\substack{i=1,\ldots,N \\ d_i=0}} L_y(G_y(G_f(x_i;\theta_f);\theta_y),y_i) - \alpha \sum_{i=1,\ldots,N} L_d(G_d(G_f(x_i;\theta_f);\theta_d),y_i). \quad (11)$$

Here, $L_y$ is the error of the category predictor, and $L_d$ is the error of domain classification. $\theta_f$ is the parameter of the feature acquisition layer. The parameter of the category predictor is $\theta_y.\theta_d$ is the parameter of the domain classifier. During the training process, to find the features with the best domain invariance, on the one hand, it is necessary to find $\theta_f$ and $\theta_y$ to minimize the category prediction error. On the other hand, it is also necessary to

search $\theta_d$ to maximize the error of domain classification.

$$(\hat{\theta}_f, \hat{\theta}_y) = \underset{\theta_f, \theta_y}{\arg\min} L(\theta_f, \theta_y, \hat{\theta}_d) \tag{12}$$

$$(\hat{\theta}_d) = \underset{\theta_d}{\arg\max} L(\hat{\theta}_f, \hat{\theta}_y, \theta_d). \tag{13}$$

Judging from the above two optimization formulas, this is a minimax problem. To solve this problem, a gradient reversal layer (GRL) is introduced into the DANN. During the process of forward propagation, the GRL acts as an identity transformation. However, during the back propagation process, the GRL automatically inverts the gradient. The optimization function selected by the DANN is a stochastic gradient descent (SGD) function. The GRL layer is generally placed between the feature extraction layer and the domain classifier layer.

The original DANN was the first proposed TL method based on adversarial networks. It is not only a method but also a general framework. Based on these foundations, many people have proposed different architectures (*Tzeng et al., 2017*; *Shen et al., 2018*; *Meng et al., 2017*).

**BGRU-based deep domain adaptation**

To process the time series data, we construct the BGRU-DANN model, the process of which is shown in Fig. 5. Source domain data and target domain data with only domain information are used to train the network. Similar to the DANN network, the BGRU-DANN network can also be divided into three parts. The first part is a feature extraction network. We use a BGRU to map the input data to a hidden state. Then, the output of the BGRU is embedded in the feature space. That is, $f = G_f(BGRU(X_k), \theta_f)$. The second part maps the new features to the label data (source domain) through the fully connected (FC) layer. That is, $\hat{y} = G_y(f, \theta_y)$. In the third part, the same feature is mapped to the domain label through the FC layer, *i.e.*, $\hat{d} = G_d(f, \theta_d)$. $G_f$ consists of a three-layer BGRU and an FC layer. A nonlinear high-dimensional feature representation of the original data is learned through the BGRU and FC layers. $G_y$ is composed of FC layers, batch normalization (BN) layers, and a rectified linear unit (ReLU) layer; $G_y$ provides the regression value of the source domain data. The network form of $G_y$ is FC1+ BN1+ ReLU1+ Dropout1+ FC2+ BN2+ ReLU2+ FC3.

During the adversarial training process, $G_d$ is used to distinguish whether the observed feature comes from the source domain or the target domain. $G_d$ consists of a gradient reversal layer and three FC layers. Here, $G_f$ is trained to extract features so that the difference between the source domain and the target domain is maximized. The labels of the source domain and target domain are set to 1 and 0, respectively. The loss function of the training process is as follows:

$$L(\theta_f, \theta_y, \theta_d) = \frac{1}{n_s}\sum_{i=1}^{n_s} L_y^i(\theta_f, \theta_y) - \alpha(\frac{1}{n_s}\sum_{i=1}^{n_s} L_d^i(\theta_f, \theta_d) + \frac{1}{n_t}\sum_{i=1}^{n_t} L_d^i(\theta_f, \theta_d)) \tag{14}$$

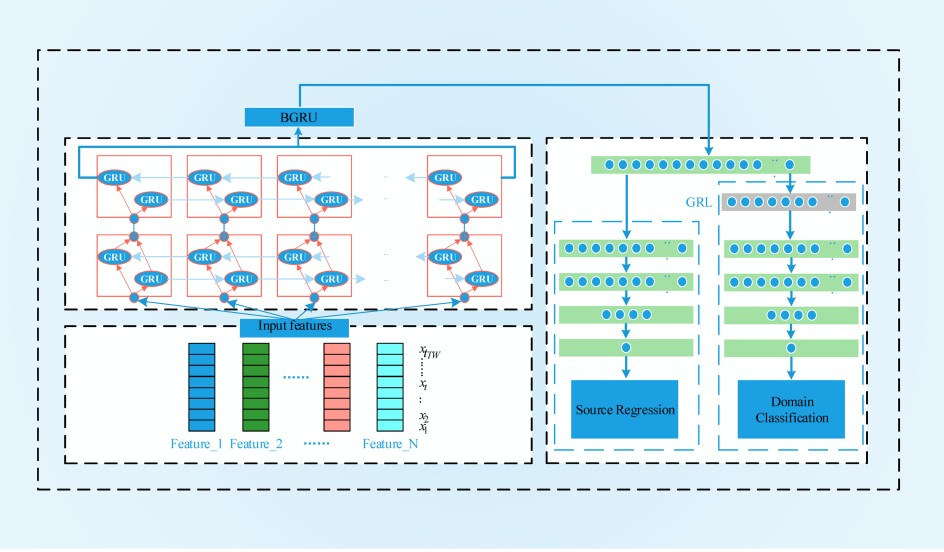

**Figure 5** **The flowchart of BGRU-DANN.**

Here, the loss functions $L_y^i$ and $L_d^i$ are defined as:

$$L_y^i(\theta_f, \theta_y) = |\hat{y}_t^i - y_t^i|^p \tag{15}$$

$$L_d^i(\theta_d, \theta_y) = d^i \log \frac{1}{\hat{d}_t^i} + (1 - d^i) \log \frac{1}{1 - \hat{d}_t^i}. \tag{16}$$

In the formula, $\hat{y}_t^i$ is the predicted value of the RUL at time $t$, *i.e.*, $\hat{y}_t^i = G_y(f_t^i, \theta_y). d^i$ is the field forecast, and $d^i = G_d(f_t^i, \theta_d). L_y^i(\theta_f, \theta_y)$ is the regression error. When the value of $p$ is different, different calculation methods can be used. $L_d^i(\theta_d, \theta_y)$ is the binary cross-loss quotient between the domain labels. The optimization process is shown in Eqs. (12) and (13). The weight update process is as follows:

$$\theta_f \leftarrow \theta_f - \lambda \left( \frac{\partial L_y^i}{\partial \theta_f} - \alpha \frac{\partial L_d^i}{\partial \theta_f} \right) \tag{17}$$

$$\theta_y \leftarrow \theta_y - \lambda \frac{\partial L_y^i}{\partial \theta_f} \tag{18}$$

$$\theta_d \leftarrow \theta_d - \lambda \alpha \frac{\partial L_d^i}{\partial \theta_f}. \tag{19}$$

Similar to a DANN, the GRL mechanism is also introduced here to realize the optimization process. SGD is used to update Eqs. (17), (18) and (19).

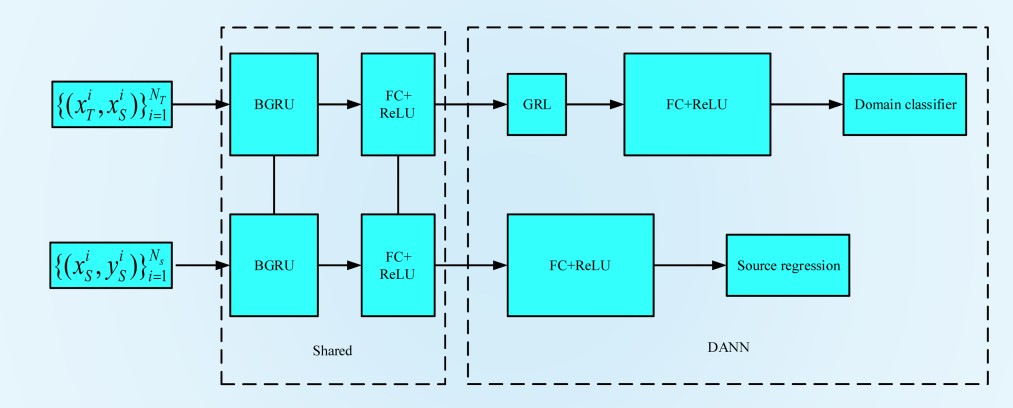

**Figure 6** **BGRU-DANN Structure.**

## BGRU-DANN structure

The structure of BGRU-DANN is shown in Fig. 6. Its basic composition can be divided into two parts. One part uses the training data from the source domain to minimize the loss of source domain regression. The other part uses the sensor data of the source domain and the target domain to maximize the error of domain classification. The BGRU and FC layers are shared by both parts. To facilitate the parameter setting process, we set the learning rates of the two sections to the same value. At the same time, we use dropout and BN layers for feature acquisition, domain classification and source domain regression. In the source domain regression task, the purpose of the training process is to minimize the regression loss function. In the domain classification task, a GRL is placed between the feature extraction and domain classification layers. During the process of back propagation, the GRL inverts the corresponding gradient to realize the optimization process of the model. When the output of the system does not improve significantly, the training process is stopped. For the corresponding FC layer, we use the ReLU activation function.

## RESULTS

### Transfer prediction

To realize the prediction of RUL, we need to establish the BGRU-DANN structure and set the corresponding hyperparameters. For different transfer tasks, the optimal parameters of the model may vary. The model in this paper has no specific optimization process for parameter setting during use, and the parameters used are the same for different transfer tasks. The input size of the BGRU network is set to 64. The size of each hidden layer is set to 256. The number of network layers is set to 3. The DANN classifier is set to a 3-layer FC structure, and the domain classifier is a 3-layer FC structure. The network learning rate is set to 0.01. The number of training iterations is set to 5000. Some of the remaining hyperparameter settings are provided in Table 5.

**Table 5  Hyperparameter settings.**

| BGRU Layers, (Units), [Dropout] | F (Units) | Source Regression Layers, (Units), [Dropout] | Domain Classification Layers, (Units), [Dropout] | $\alpha$ | $\lambda$ |
|---|---|---|---|---|---|
| 3, (64, 256), [0.5] | (256) | 3, (256, 128, 32), [0.5] | 3, (256, 128, 32), [0.5] | 0.5 | 0.01 |

After setting the relevant parameters, we can predict the RUL. First, we use the BGRU structure to extract the features of the input sequence data. Then, the DANN network is used to implement adversarial training to extract features with domain invariance. The experimental results are shown in Figs. 7 and 8.

Figure 7 reflects the predicted results of bearing 2-1, bearing 2-4 and bearing 2-6. The source domain data are bearing 1-3-bearing 1-7, and the target data are bearing 2-1, bearing 2-4, and bearing 2-6. Figure 8 reflects the prediction results for bearing 3-1, bearing 3-2, and bearing 3-3. The source domain data are bearing 1-3-bearing 1-7, and the target data are bearing 3-1, bearing 3-2, and bearing 3-3. From (A), (C), and (E) in Fig. 7 and (A), (C), and (E) in Fig. 8, we can conclude that the predicted RUL results exhibit a good downward trend performance and are very close to the real RUL values; this effectively illustrates the effectiveness of the proposed data-driven prediction framework based on TL.

## Comparison of experimental results

To demonstrate the advantages of data-driven prediction methods based on domain adaptation, three methods are used for comparison purposes, namely, a BGRU without transfer learning, TCA-NN, and FC-DANN.

We can see in Figs. 7 and 8 that the RUL prediction results of BGRU-DANN are significantly better than those of the other three methods, and the declining trend can best reflect the real RUL value. However, the other three methods cannot reflect the degradation trend of the RUL effectively.

Figure 9 shows the RUL errors of BGRU-DANN, the BGRU, TCA-NN and FC-DANN. It can be clearly seen from Fig. 9 that the RUL error generated by the BGRU-DANN model is the smallest, especially for bearings 2-4, 3-1 and 3-3. At the same time, bearing 2-1 and bearing 2-6 in Fig. 9 clearly reflect that the RUL error generated by BGRU-DANN is smaller than that of the other three methods in most cases. Bearing 3-2 in Fig. 9 may not clearly indicate the superiority of BGRU-DANN due to the large amount of data involved. However, through the comparison of the three evaluation indicators in Table 6, it can still be seen that BGRU-DANN achieves the best effect.

Table 6 shows that BGRU-DANN achieves the best results in terms of the three evaluations, the MAE, MSE, and R2_score, which further proves the effectiveness of the method proposed in this paper. Regarding the MSE, the calculated results of the proposed method for bearing 2-1, bearing 2-4, bearing 2-6, bearing 3-1, bearing 3-2 and bearing 3-3 are 0.0283, 0.0193, 0.0217, 0.0298, 0.0503, and 0.0472, respectively, which are far less than the calculated error results of the other three methods. For the MAE, the calculated results of the proposed method for bearing 2-1, bearing 2-4, bearing 2-6, bearing 3-1, bearing 3-2 and bearing 3-3 are 0.1157, 0.0928, 0.0875, 0.1215, 0.1569, and 0.1238, respectively, which are still better than the calculated error results of the other three

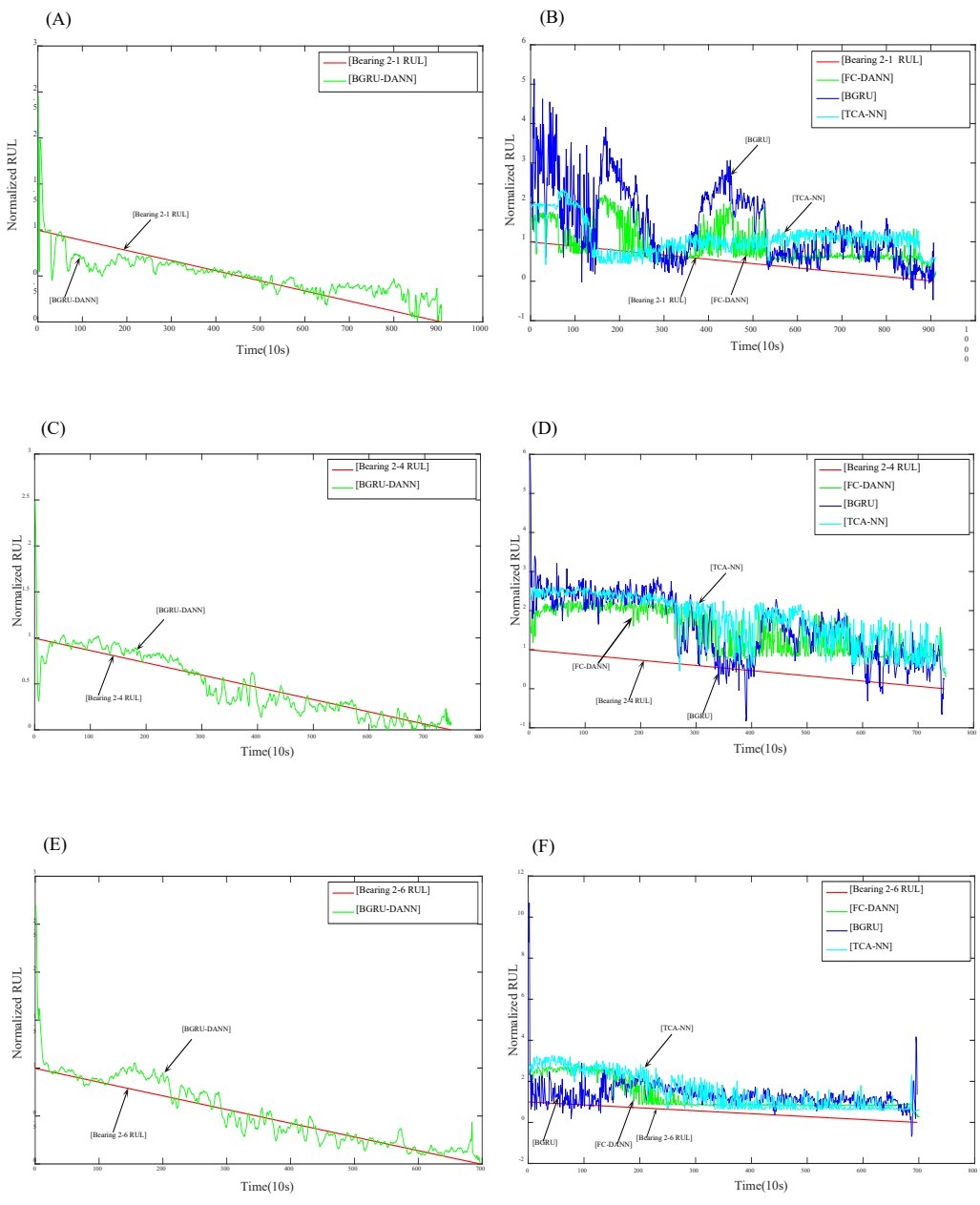

**Figure 7  Prediction results for dataset 2.** (A) Prediction results of Bearing 2-1 using BGRU-DANN; (B) prediction results of Bearing 2-1 using the comparison method; (C) prediction results of Bearing 2-4 using BGRU-DANN; (D) prediction results of Bearing 2-4 using the comparison method; (E) prediction results of Bearing 2-6 using BGRU-DANN; (F) prediction results of Bearing 2-6 using the comparison method.

models. In terms of the R2-score calculation results, the calculated results of the proposed method for bearing 2-1, bearing 2-4, bearing 2-6, bearing 3-1, bearing 3-2 and bearing 3-3 are 0.6576, 0.7664, 0.7367, 0.6379, 0.3935, and 0.4252, respectively; this indicates that the model has certain explanatory ability regarding the relationship between the independent

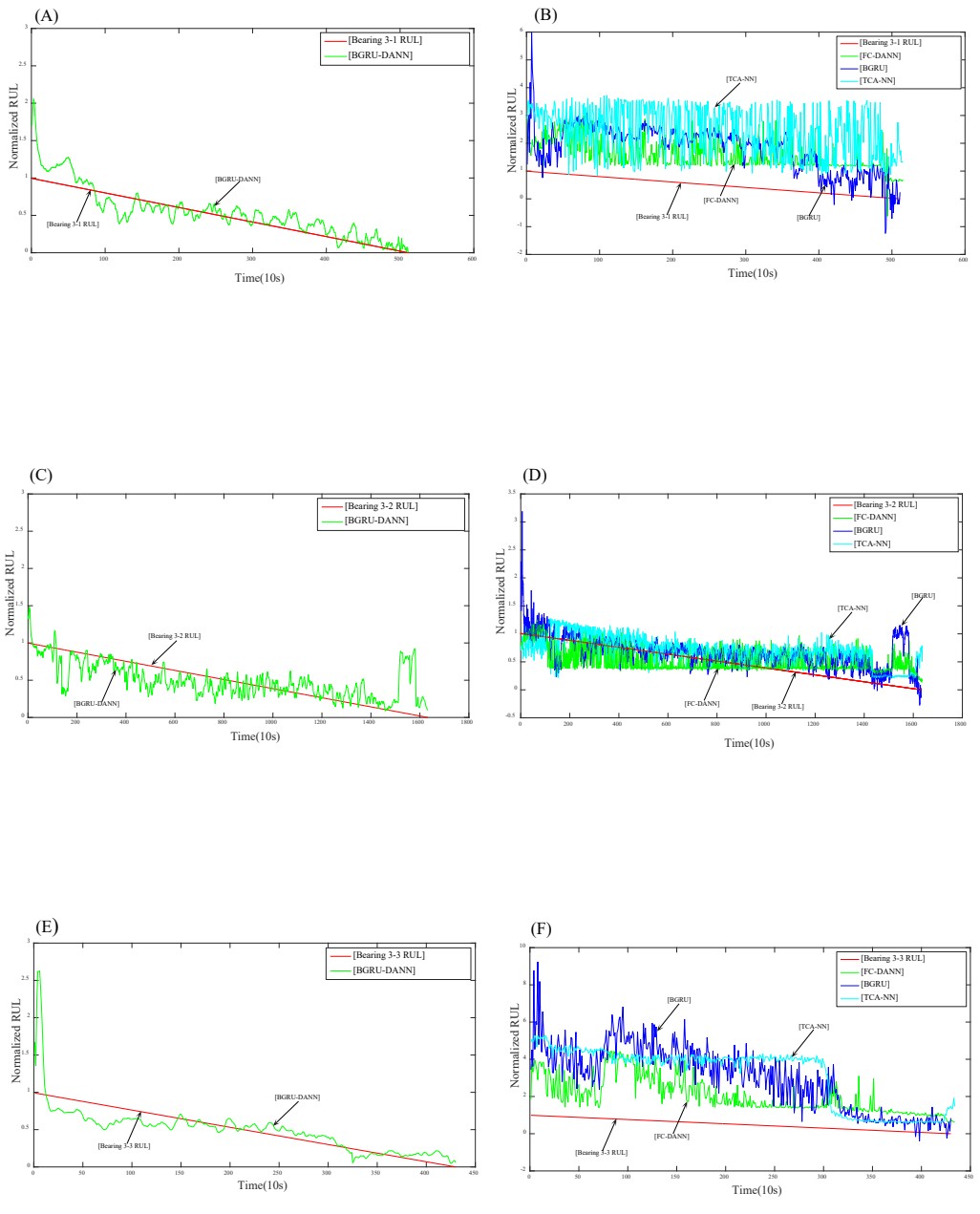

**Figure 8  Prediction results for dataset 3.** (A) Prediction results of Bearing 3-1 using BGRU-DANN; (B) prediction results of Bearing 3-1 using the comparison method; (C) prediction results of Bearing 3-2 using BGRU-DANN; (D) prediction results of Bearing 3-2 using the comparison method; (E) prediction results of Bearing 3-3 using BGRU-DANN; (F) prediction results of Bearing 3-3 using the comparison method.

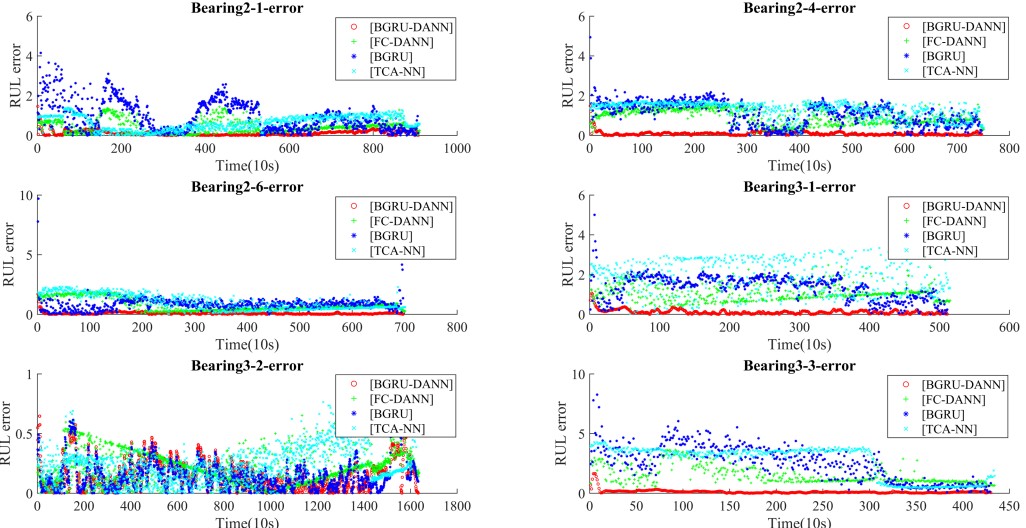

**Figure 9** RUL error.

**Table 6** Performance metrics for the datasets.

| Dataset | Performance metric | The proposed method | BGRU | TCA-NN | FC-DANN |
|---|---|---|---|---|---|
| bearing 2-1 |  | 0.0283 | 1.4652 | 0.5205 | 0.2865 |
| bearing 2-4 |  | 0.0193 | 1.5442 | 1.8181 | 1.0214 |
| bearing 2-6 | MSE | 0.0217 | 1.0589 | 1.4436 | 0.9164 |
| bearing 3-1 |  | 0.0298 | 2.1957 | 4.4414 | 1.3557 |
| bearing 3-2 |  | 0.0503 | 0.0883 | 0.0796 | 0.0606 |
| bearing 3-3 |  | 0.0472 | 8.9927 | 8.8676 | 2.9564 |
| bearing 2-1 |  | 0.1157 | 0.9440 | 0.6316 | 0.4218 |
| bearing 2-4 |  | 0.0928 | 1.0940 | 1.2917 | 0.9429 |
| bearing 2-6 | MAE | 0.0875 | 0.8491 | 1.0226 | 0.8063 |
| bearing 3-1 |  | 0.1215 | 1.3532 | 1.9420 | 1.0884 |
| bearing 3-2 |  | 0.1569 | 0.2070 | 0.2377 | 0.2035 |
| bearing 3-3 |  | 0.1238 | 2.5813 | 2.6589 | 1.5394 |
| bearing 2-1 |  | 0.6576 | −16.6992 | −5.2325 | −2.4311 |
| bearing 2-4 |  | 0.7664 | −17.6799 | −20.7599 | −11.2249 |
| bearing 2-6 | R2_score | 0.7367 | −11.8176 | −16.2749 | −9.9658 |
| bearing 3-1 |  | 0.6379 | −25.6589 | −52.0910 | −15.2054 |
| bearing 3-2 |  | 0.3935 | −0.06367 | 0.0451 | 0.2733 |
| bearing 3-3 |  | 0.4252 | −108.42439 | −104.9228 | −34.3136 |

variable and the dependent variable in the regression analysis and is superior to the three compared methods.

## CONCLUSIONS

In this article, a domain-adaptative prediction method based on deep learning with a BGRU and a DANN is proposed. The validity of the proposed method is demonstrated by an experiment on the 2012 IEEE PHM dataset. The objective of this study is to propose a domain-adaptive RUL prediction method. When the input bearing is transferred from the source domain with label information to a target domain with only sensor information, a more accurate estimate of the RUL can be obtained. From the results of the experiment, we can draw the following conclusions:

(1) Compared with the BGRU without TL, the proposed method has a better effect in terms of RUL prediction. This indicates that the model obtained by adversarial training has better generalization ability and can adapt to data with different distributions.

(2) The comparison with TCA-NN proves that the deep, domain-adaptive BGRU-DANN method has better performance. This indicates that the transfer method based on deep learning has a stronger feature extraction ability than the traditional nondeep transfer method, and it can extract better features with domain invariance.

(3) Using FC layers for feature extraction, this paper constructs an FC-DANN network. A comparison of the results fully shows that the BGRU has a better effect in terms of feature extraction. Compared with the features extracted by the FC method, the features extracted by the BGRU for sequence data processing are more representative.

(4) By means of domain adaptation, the generalization ability of the data-driven RUL prediction model can be effectively improved, and it can adapt to RUL prediction tasks under different working conditions to a certain extent.

In future work, we will take a closer look at the problem of time series transfer. Remaining life prediction problems with respect to bearings, aero engines, etc. can actually be regarded as time series transfer problems. However, research on time series transfer is still in its infancy. There are merely a few studies on such issues. Only Ye proposed two different time series transfer methods in references (*Ye & Dai, 2018*; *Ye & Dai, 2021*), one based on an extreme learning machine and the other based on a CNN. However, most of the data monitored by sensors are time series data, and this is a very common data type in RUL forecasting research. Therefore, the authors intend to conduct related research in the future, hoping to obtain a better model and research results with more practical application value.

### Funding

The authors received no funding for this work.

### Competing Interests

The authors declare there are no competing interests.

## Author Contributions

- Bin cheng Wen conceived and designed the experiments, performed the computation work, prepared figures and/or tables, and approved the final draft.
- Ming qing Xiao performed the experiments, authored or reviewed drafts of the paper, and approved the final draft.
- Xue qi Wang performed the experiments, prepared figures and/or tables, and approved the final draft.
- Xin Zhao analyzed the data, authored or reviewed drafts of the paper, and approved the final draft.
- Jian feng Li performed the computation work, authored or reviewed drafts of the paper, and approved the final draft.
- Xin Chen analyzed the data, prepared figures and/or tables, and approved the final draft.

## Data Availability

Data and code are available on at GitHub: https://github.com/WEN-AFK/BGRU-DANN. The IEEE PHM 2012 Data Challenge Dataset is located in Full_Test_Set, Learning_Set, and Test_Sst.

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
