# Peer review of "Data-driven remaining useful life prediction based on domain adaptation"

_PeerJ Computer Science, doi:10.7717/peerj-cs.690_

## Round 0.1 · original submission · Major Revisions

The paper should be modified based on the review comments. I encourage you to resubmit after making these corrections.

Reviewer 1 ·

Basic reporting

1. The paper has some grammatical errors and proofreading of the paper needs to be done.
2. Relevant references in line number 47 are not cited for the RNN techniques used popularly
3. In the introduction section, the authors discuss the merits of using deep learning techniques under data-driven models for RUL predictions and the demerits of Machine learning techniques(line no 49). But no discussion is provided on the popular machine learning techniques such as SVM, Naive Bayes etc
4. The authors can also discuss the use of other RNN architectures such as BRNN or Bayesian RNN for RUL prediction
5. The authors highlight four categories of transfer learning- Instance-based Transfer Learning, Feature-based Transfer Learning, Model-based Transfer Learning and Relation based Transfer Learning. However, which category they have adopted in their paper is not mentioned.
6. In Literature Review section, techniques with references are only cited but detailed explanation of only a few research work is given. Authors should add more detailed literature.

Experimental design

1. An extensive diagram depicting the entire system methodology need to be added.
2. Explanation for using sliding window is not provided
3. Author states that feature vector is formed using extracted feature. Detail explanation is need on the same like the vector size formed etc.
4. Also explanation regarding the input of features to the model is missing.
5. The figure quality of figure 2,5 and 6 needs to be improved. The text in some of the figures are not visible.

Validity of the findings

1. Result section explanation need to be improved. Result needs to be explained in terms of accuracy or error.
2. Similarly the conclusion section need to be explained point wise based on model performance and comparison with other models used.

·

Basic reporting

It is assumed that the training data and test data fit the same distribution. This article proposes a data-driven framework with domain adaptability using a two-way gated repeating unit (BGRU). The framework used the domain opposite neural network (DANN) to implement the transfer from the source domain to the destination domain that contains only sensor information. We analyzed the IEEE PHM 2012 Challenge datasets to verify the effectiveness of the method and used them for validation.
The article is written in clear and understandable language. Literature references are sufficient. The article age and figures are given appropriately. However, the visibility of the visuals needs to be updated. I think some corrections and additions need to be made. My suggestions about the article are listed below.

Experimental design

The subject and content of the article is suitable for the content of the journal. It is considered that rigorous research is carried out according to a good technical and ethical standard.

Validity of the findings

-Explanation of the RUL expression should be given in the Abstract section, which is the first section it is used. “remaining useful life (RUL)”
-The Abstract section should be better worded. "In response to existing problems, this paper proposes a data-driven framework with domain adaptability using bidirectional gated recurrent unit (BGRU)." In this panel "existing problems" should be specified.
-The resolution of the figures can be increased.
-Labels are not read in Figures 5 and 6. Typefaces can be enlarged.
-In the Results section; The results obtained should be compared with the literature and the obtained values should be discussed in more detail.
-Conclusion section "[41], [42]" it refers instead should be made in a few sentences çneri.
The results obtained in the conclusion section should be given as items. If there are numerical results, they should be given here briefly.

Additional comments

-Explanation of the RUL expression should be given in the Abstract section, which is the first section it is used. “remaining useful life (RUL)”
-The Abstract section should be better worded. "In response to existing problems, this paper proposes a data-driven framework with domain adaptability using bidirectional gated recurrent unit (BGRU)." In this panel "existing problems" should be specified.
-The resolution of the figures can be increased.
-Labels are not read in Figures 5 and 6. Typefaces can be enlarged.
-In the Results section; The results obtained should be compared with the literature and the obtained values should be discussed in more detail.
-Conclusion section "[41], [42]" it refers instead should be made in a few sentences çneri.
The results obtained in the conclusion section should be given as items. If there are numerical results, they should be given here briefly.

---

## Round 0.2 · Minor Revisions

Thanks for your interest in the journal!

Please address the comments of Reviewer 1 under "Experimental design".

Reviewer 1 ·

Basic reporting

1. The authors have incorporated all the suggestions.

2. The grammatical errors are removed and the manuscript is now ready in professional English.

3. Literature Review is now detailed. Few reference articles relevant to the manuscript are mentioned below. The authors can choose to add them in Literature Review section:

i. Kamat, P., Marni, P., Cardoz, L., Irani, A., Gajula, A., Saha, A., ... & Sugandhi, R. (2021). Bearing Fault Detection Using Comparative Analysis of Random Forest, ANN, and Autoencoder Methods. In Communication and Intelligent Systems (pp. 157-171). Springer, Singapore.

ii. Motahari-Nezhad, M., & Jafari, S. M. (2021). Bearing remaining useful life prediction under starved lubricating condition using time domain acoustic emission signal processing. Expert Systems with Applications, 168, 114391. https://doi.org/10.1016/J.ESWA.2020.114391

iii. Wang, X., Wang, T., Ming, A., Zhang, W., Li, A., & Chu, F. (2021). Spatiotemporal non-negative projected convolutional network with bidirectional NMF and 3DCNN for remaining useful life estimation of bearings. Neurocomputing, 450, 294–310. https://doi.org/10.1016/J.NEUCOM.2021.04.048

iv. Sayyad, S., Kumar, S., Bongale, A., Bongale, A. M., & Patil, S. (2021). Estimating Remaining Useful Life in Machines Using Artificial Intelligence: A Scoping Review. Library Philosophy and Practice, 1-26.

Experimental design

1. The authors can discuss the relation of the extracted features such as kurtosis, skewness, etc with bearing life.

2. Importance of the feature extraction step can be justified.

Validity of the findings

No comment

·

Basic reporting

It is seen that the authors made the necessary arrangements. The article is acceptable as it is.

Experimental design

The article has been prepared in adequate technical and ethical standards.

Validity of the findings

The results of the article are well expressed, linked to the original research question.

Additional comments

The language of the article and the findings have been developed. The article is better worded.

---

## Round 0.3 · accepted · Accept

Thanks for your interest in the journal. We look forward to your future submissions.